# Therapeutic Effects of Urethral Sphincter Botulinum Toxin A Injection on Dysfunctional Voiding with Different Videourodynamic Characteristics in Non-Neurogenic Women

**DOI:** 10.3390/toxins13050362

**Published:** 2021-05-19

**Authors:** Yuan-Hong Jiang, Cheng-Ling Lee, Sheng-Fu Chen, Hann-Chorng Kuo

**Affiliations:** Department of Urology, Hualien Tzu Chi Hospital, Buddhist Tzu Chi Medical Foundation, Tzu Chi University, Hualien 970, Taiwan; redeemerhd@gmail.com (Y.-H.J.); leecl@hotmail.com (C.-L.L.); madaux@yahoo.com.tw (S.-F.C.)

**Keywords:** urethra, onabotulinumtoxinA, voiding, therapeutic outcome

## Abstract

Although female dysfunctional voiding (DV) is common in urological practice, it is difficult to treat. This study evaluated the therapeutic efficacy of urethral botulinum toxin A (BoNT-A) on non-neurogenic female DV. Based on the videourodynamic study (VUDS), the DV was classified into three subgroups according to the obstructive site. A successful treatment outcome was defined as an improvement of voiding efficiency by 10% and reported global response assessment by ≥1. The study compared therapeutic efficacy, baseline urodynamic parameters, and changes in urodynamic parameters between the treatment success and failure groups and among three DV subgroups. Predictive factors for successful treatment were also investigated. A total of 81 women with DV were categorized into three groups: 55 (67.9%) had mid-urethral DV, 19 (23.5%) had distal urethral DV, and 7 (8.6%) had combined BN dysfunction and mid-urethral DV after BN transurethral incision. The treatment outcome was successful for 55 (67.9%) patients and failed for 26 (32.1%). Successfully treated patients had a significant decrease of detrusor pressure, post-void residual volume, and bladder outlet obstruction index, as well as an increase in voiding efficiency at follow-up versus the treatment failure group. The logistic regression of urodynamic parameters and clinical variables revealed that a greater volume of first sensation of filling predicts a successful BoNT-A treatment outcome (*p* = 0.047). The urethral BoNT-A injection is effective in treating non-neurogenic women with DV, with a success rate of 67.9%. The videourodynamic characteristics of DV may differ among patients but does not affect the treatment outcome.

## 1. Introduction

The pelvic floor function plays an essential role in micturition and defecation [1]. During micturition, the pelvic floor muscles and external urethral sphincter should appropriately relax to facilitate sustained detrusor contraction and complete bladder emptying [2]. When the pelvic floor muscles or urethral sphincter cannot relax during micturition, it is called dysfunctional voiding (DV). The prevalence of female voiding dysfunction is 2.7% to 23% [3,4,5]. Female DV is commonly encountered in urological practice but is difficult to treat. In a large cohort of female voiding dysfunction, 17% of patients had DV and 17.6% had poor relaxation of the external sphincter [6]. Among women with clinically unsuspected bladder outlet obstruction (BOO), DV is the most common form of voiding dysfunction [7]. Since female DV might present with storage symptoms rather than voiding symptoms as their chief complaints, DV is commonly mis-diagnosed as an overactive bladder and treated [8]. The accurate diagnosis and treatment is important in this voiding dysfunction.

According to the International Continence Society (ICS) and International Urogynecological Association (IUGA) joint report, DV is characterized by an intermittent and/or fluctuating flow rate due to involuntary intermittent contractions of the peri-urethral striated or levator muscles during voiding in neurologically normal women. This type of voiding may also be the result of an acontractile detrusor (abdominal voiding) with electromyography (EMG) or video-urodynamics (VUDS) required to distinguish between the two entities [9]. The VUDS of DV is characterized by increased external sphincter activity during the voiding phase, resulting in high voiding detrusor pressure (Pdet), low maximum flow rate (Qmax), and increased post-void residual (PVR) urine volume, as well as a typical downward spiral shape in voiding cystourethrography on VUDS [4,6,7,8,9,10].

Many treatment modalities have been used to relax the urethral sphincter and pelvic floor muscles, including pelvic floor muscle training, antimuscarinic therapy, sacral nerve neuromodulation, and posterior tibial nerve stimulation [5,11,12]. Since no definitive medical treatment is currently available for DV, clinicians have enthusiastically used urethral botulinum toxin A (BoNT-A) for this off-label indication [13]. However, a satisfactory treatment outcome with BoNT-A has not been achievable for all patients with DV [14].

In VUDS of female DV, different characteristics are observed in the urethral obstruction, such as a typical downward spiral shape mid-urethra, a narrow distal urethra, and a narrow mid-urethra after incision of previous bladder neck (BN) obstruction. These different VUDS characteristics suggest a different underlying pathophysiology of female DV [15,16]. This study retrospectively evaluated the therapeutic efficacy of urethral BoNT-A on non-neurogenic female DV and searched for predictive factors for a successful treatment outcome.

## 2. Results

Participants were 81 women with DV. Among these 81 patients, 55 (67.9%) had a successful outcome and 26 (32.1%) experienced treatment failure. The urodynamic parameters between groups are listed in Table 1. No significant difference was noted between the treatment success and failure groups in age distribution, urodynamic parameters, presence of central nervous system lesions, and VUDS subtypes of DV.

Based on the VUDS characteristics of DV, patients were categorized into one of three groups: 55 (67.9%) had mid-urethral DV and obstruction; 19 (23.5%) had distal urethral DV and obstruction, and 7 (8.6%) had combined BN dysfunction and DV with mid-urethral obstruction present after transurethral incision of-BN. Table 2 lists the baseline VUDS parameters of these DV subgroups. Patients with distal DV were significantly older than the other two groups, but the mean Pdet at Qmax (Pdet.Qmax) was significantly lower among three subgroups. Patients with BN dysfunction plus DV had the highest Pdet and BOOI. An excellent result was reported in 15 (18.5%) patients, markedly improved in 25 (30.9%) and mildly improved in 15 (18.5%). However, the treatment success rate was similar among three groups.

The changes of lower urinary tract symptoms (LUTS) at baseline and after urethral BoNT-A injection are shown in Table 3. In overall patients, storage LUTS improved from 70.4% to 50.6%, voiding LUTS improved from 87.7% to 42%, and painful LUTS improved from 13.3% to 4.9%. Patients with BND plus DV had greater improvement in voiding LUTS (−57.1%) and patients with mid-urethral DV had less improvement in storage LUTS (−18.2%) among three DV subgroups.

After urethral BoNT-A injections, a significant decrease of Pdet.Qmax, PVR volume, cystometric bladder capacity, and BOOI and an increase of VE were noted in patients with a successful treatment outcome, but not in the treatment failure group (Table 4). Patients with DO during the filling phase showed no significant change in the Pdet of DO after urethral BoNT-A injection. The rates of medical comorbidities such as diabetes mellitus, hypertension, cardiac arterial disease, chronic kidney disease, chronic obstructive pulmonary disease, congestive heart failure, and central nervous system lesions did not significantly differ among groups. Logistic regression of the urodynamic parameters, including DO subtypes, and clinical variables revealed that a greater first sensation of filling predicts a successful treatment outcome (*p* = 0.047).

Adverse events after urethral BoNT-A injection include hematuria (5/81, 6.2%) and difficulty in urination (7/81, 8.6%). These adverse events were mild and resolved within 3 days after the treatment. No systemic adverse events were reported.

## 3. Discussion

This study demonstrated that urethral BoNT-A injection is effective in treating female non-neurogenic DV with a successful rate of 67.9%. Patients with a successful outcome had a significant decrease of Pdet.Qmax, an increase of voiding efficiency (VE), and a decrease of bladder outlet obstruction index (BOOI). However, the study failed to find the difference of the successful treatment outcome among different VUDS characteristics of DV in this cohort.

Poor relaxation of the urethral sphincter has long been considered a cause of female voiding dysfunction [2,17]. The term “urethral syndrome” has been used to describe a spastic urethral sphincter during micturition. However, the muscles contributing to DV in women involve the urethral sphincter and pubococcygeal muscles, causing BOO and voiding dysfunction in women [18]. Dysregulated urethral function with spastic or a nonrelaxing external urethral sphincter is thought to be a possible cause of DV and results in voiding symptoms, slow urinary flow, and large PVR [16,17,18]. However, the actual mechanism for DV is not well understood. Therefore, the attempt to reduce urethral sphincter hypertonicity by BoNT-A injection and resume spontaneous voiding usually cannot achieve a high satisfactory rate. When BoNT-A is injected into the urethral sphincter, the BoNT-A solution may not have an adequate effect on the urethral sphincter and pubococcygeal muscles, causing inhomogeneous therapeutic effects. As shown in Table 4, the Pdet.Qmax in the treatment failure group was also reduced after BoNT-A injections, although the difference did not reach significance. This result also implies that, although BoNT-A still has an influence on relaxing the urethral sphincter, the effect may be insufficient to resume spontaneous and efficient voiding, resulting in a low VE, large PVR, and low Qmax after treatment.

Urethral sphincter BoNT-A injection has been used to treat male and female voiding dysfunction for decades [19]. The indications include detrusor underactivity, neurogenic detrusor sphincter dyssynergia, and non-neurogenic DV. Although most studies demonstrated the efficacy of this treatment in voiding dysfunction caused by urethral sphincter hyperactivity of different etiologies, the subjective improvement rate after BoNT-A urethral sphincter injection was only 60% [13,14,19]. In the experience of the authors for the present study, patients who have an open BN on voiding cystourethrography have predictably successful therapeutic results [14]. In this study, women with DV that already had a wide open BN during voiding had the greater improvement of voiding LUTS, however, their urethral sphincter or pelvic floor muscle still could not relax appropriately for a spontaneous voiding in all patients, which suggests that the pathophysiology of DV may be more complex than originally understood.

Based on the VUDS characteristics in this study, three DV subtypes were found: (1) Combined BN obstruction and DV; (2) mid-urethral obstruction with high Pdet.Qmax; and (3) distal urethral obstruction with high Pdet.Qmax. In the seven patients who had both BN obstruction and DV, the mid-urethral obstruction persisted after transurethral incision-BN, suggesting that the increased sympathetic hyperactivity results in BN dysfunction and increased urethral smooth muscle hypertonicity might play a role in female DV. However, urethral striated muscle spasm due to increased pudendal hyperactivity may also coexist. For patients with mid-urethral obstruction DV, the chronic sphincteric spasm may contribute to the changes of bladder dynamics and morphology [20]. Regarding the distal urethral obstruction DV, the pelvic floor muscle such as pubococcygeal muscles may prohibit complete dilatation of the urethra during voiding, resulting in a urethral structure-like BOO during voiding [18].

Although the VUDS characteristics of female DV are distinct, the treatment outcome of BoNT-A injection did not differ among subgroups. A relatively higher success rate was noted in patients with mid-urethral DV (72.7%), indicating that the cause of this DV subtype may be the urethral striated sphincter spasticity and that the voiding would become less difficult after BoNT-A urethral injections. For the other subtypes, urethral BoNT-A injection to the urethral sphincter alone may not ameliorate the pathophysiology of DV that involves urethral smooth muscle or the pelvic floor muscles. Further pelvic floor muscle training for relaxation or medication to relax the urethral smooth muscle may be helpful in this case [21,22].

A major cause of female DV is DO that causes volitional sphincteric activity to overcome unanticipated detrusor contractions during storage [6,8]. Therefore, anticholinergic medication has been widely used as a primary modality to treat DO associated with DV [23]. The present study also found that a high percentage of patients with DV had urodynamic DO, and 13 of 15 patients with a central nervous system lesion and DO had a successful treatment outcome, suggesting that the urethral sphincter hyperactivity may be secondary to uninhibited detrusor contractions. Urethral BoNT-A injection relaxes the urethral sphincter and improves the voiding condition. However, most women with DV had both storage and voiding symptoms, and the presence of DO and storage symptoms did not predict a successful treatment outcome with BoNT-A. These results suggest DO as an etiology of female DV. Therefore, a longer treatment period for DO may improve the success rate after urethral BoNT-A injections.

The treatment outcome of female DV has not been well documented, possibly because the definition of DV has not been fully established. This study used a strict definition for DV, including a high Pdet.Qmax, a low Qmax, and a narrowing urethra on VUDS. However, this definition may exclude the women with voiding dysfunction due to poor relaxation of the pelvic floor muscles [24]. A recent study by the same authors also demonstrated that a more prominent obstructive profile at baseline VUDS in female DV was associated with a higher rate of BOOI response at follow-up [25]. Women who do not respond to urethral BoNT-A injections may have a less obstructive urethral condition or the cause of their voiding dysfunction was not at the urethral sphincter. Confirmation of the obstructive site of DV is important in selecting female patients for whom urethral BoNT-A injections are suitable.

Although this study did not find difference of successful BoNT-A treatment outcome among different VUDS subgroups, Pdet.Qmax decreased both in the treatment success and failure subgroups after BoNT-A injections. The PVR and BOOI also decreased, and VE increased in the treatment success subgroup but not the failure subgroup. These urodynamic results indicate that urethral BoNT-A injection did relax the urethral sphincter and decrease the urethral resistance during voiding. Interestingly, a Qmax increase was not observed after successful BoNT-A injection. Voiding is a complex interaction of the central nervous system and peripheral neural control, including sustainable detrusor contractions, adequate BN relaxation, and complete external sphincter relaxation and pelvic floor relaxation [26]. Patients with DV usually have impaired VE due to involuntary urethral sphincter hyperactivity and inhibiting sustained detrusor contractions. Non-neurogenic voiding dysfunction may be due to an enhanced guarding reflex, causing a nonrelaxing urethral sphincter, significant PVR or urinary retention [27]. Although BoNT-A injections relax the urethral sphincter, the therapeutic effect of 100 U BoNT-A may not be sufficiently strong to reverse the guarding reflex and resume normal detrusor contractility. Increasing the dose of BoNT-A or repeating BoNT-A injections may improve the success rates and efficacy of treatment.

## 4. Conclusions

The urethral sphincter BoNT-A injection is effective in treating female non-neurogenic DV with a success rate of 67.9%. Patients with a successful treatment outcome had improved Pdet.Qmax, increased VE, and decreased PVR and BOOI. Three subgroups were characterized based on VUDS features in DV, but the success rates were similar among subgroups, suggesting that the pathologic mechanisms of female DV are more complicated than currently understood.

## 5. Materials and Methods

This retrospective analysis reviewed all women with DV refractory to medical treatment who received urethral BoNT-A 100 U (onabotulinumtoxinA, Allergan, Irvine, CA, USA) injection at the authors’ hospital. The diagnosis of DV was made by VUDS under fluoroscopy and multichannel urodynamic equipment. The definition of DV was in accordance with that recommended by the ICS and IUGA joint report [9]. All patients included in this study had evident radiographic obstruction at the middle urethra with an open BN and characteristic urodynamic findings such as high Pdet.Qmax and low Qmax. This study was approved by the Institutional Review Board of the authors’ hospital (IRB: 105-151-B, approval date 29 December, 2016). Informed consent was waived due to the retrospective study design.

All patients underwent VUDS and cystoscopy before the BoNT-A injections to confirm that no urethral stricture or anatomic BOO were present. The LUTS including storage (urgency/frequency/nocturia, urgency urinary incontinence) LUTS, voiding (difficult urination, urinary retention) LUTS, and painful (bladder pain, micturition pain) LUTS were recorded before and 1 to 3 months after urethral BoNT-A injection. All patients were informed of the possible adverse events after BoNT-A injection, including exacerbation of urgency urinary incontinence and mild miction pain.

Based on the recommendation of the ICS/IUGA joint report, VUDS was performed as a standard procedure [9]. During the VUDS examination, patients were placed in a supine position. A dual-channel urethral catheter was inserted, the PVR was evacuated, and an 8-Fr rectal catheter and surface electromyographic leads were taped in the perineal area. After the pressure recording was calibrated and the intravesical and intra-abdominal pressures were balanced, patients were moved to a sitting position, with the uroflowmeter placed under the commode. A C-arm fluoroscope was positioned at a 45-degree angle below the buttocks. After proper adjustment of the C-arm position, filling cystometry at a filling rate of 30 mL/min, a voiding pressure flow study, and concomitant fluoroscopic voiding cystourethrography were performed with 20% Urografin in normal saline [8].

These VUDS parameters were recorded and analyzed: First bladder sensation, fullness sensation, urge sensation, cystometric bladder capacity, Pdet.Qmax, Qmax, PVR, and abdominal pressure to void. The terminology used in this study aligned with the International Continence Society recommendations. Detrusor overactivity (DO) during the filling phase and before the uninhibited voiding detrusor contraction were recorded as phasic DO and terminal DO, respectively. The VE, BOOI (defined as Pdet–2 × Qmax) were calculated from the measured parameters. During voiding, the bladder outlet appearance was carefully evaluated. Urethral narrowing was defined as BOO at the mid-urethra or distal urethra (Figure 1). Some patients who had previous transurethral incision for BN dysfunction but were found to have mid-urethral BOO during follow-up were specifically categorized as having combined BN dysfunction and DV.

Urethral sphincter BoNT-A injections were performed in the operation room with the patient under light intravenous general anesthesia [19]. Each vial of 100 U BoNT-A was reconstituted to 4 mL with normal saline to create a concentration of BoNT-A solution equivalent to 25 U/mL. The BoNT-A dose was 100 U for all patients. Cystoscopy was performed before the BoNT-A injection to determine the urethra axis for the proper injection position. Using a 23-gauge, 1-mL syringe, BoNT-A was injected into the urethral sphincter along the urethral lumen at the 2, 5, 7, 9, and 12 o’clock positions of the sides of the urethral meatus. The effect of BoNT-A typically appeared within 1 week after the urethral sphincter injections, and the maximum effect was reached at about 2 weeks after injection [19]. Antibiotics were given for 3 days to prevent urinary tract infections due to the urethral injection. Medications for reduction of urethral resistance were discontinued after the BoNT-A injections. Patients who were taking antimuscarinics for the overactive bladder symptoms continued the medication after the urethral BoNT-A injection. No concomitant intra-detrusor BoNT-A injection was performed.

The BoNT-A treatment outcome was assessed at 1–3 months after the urethral BoNT-A injection. Patients were requested to report their global response assessment as excellent (+3), markedly improved (+2), mildly improved (+1), no change (0) or worsened (−1), according to their perception of voiding after the BoNT-A injection. This study only enrolled patients with available baseline and postoperative urodynamic data. Patients with an improvement of VE by 10% and global response assessment by ≥1 were considered to have a successful treatment outcome, otherwise the treatment was considered a failure. Adverse events after the BoNT-A injections were recorded and appropriately treated.

Continuous variables were expressed as the mean and standard deviation, and categorical data were presented as the number and percentage. The chi-square test for categorical variables and the Wilcoxon rank-sum test for continuous variables were used to determine the p-values between groups for statistical comparisons. All statistical assessments were two-sided and considered significant at *p* < 0.05. All calculations were performed using SPSS for Windows (version 16.0, Chicago, IL, USA). The therapeutic efficacy, baseline urodynamic parameters, and changes of urodynamic parameters between successful and failure groups were compared, and the logistic regression analysis was performed to find the predictive factor for a successful treatment outcome.

## Figures and Tables

**Figure 1 toxins-13-00362-f001:**
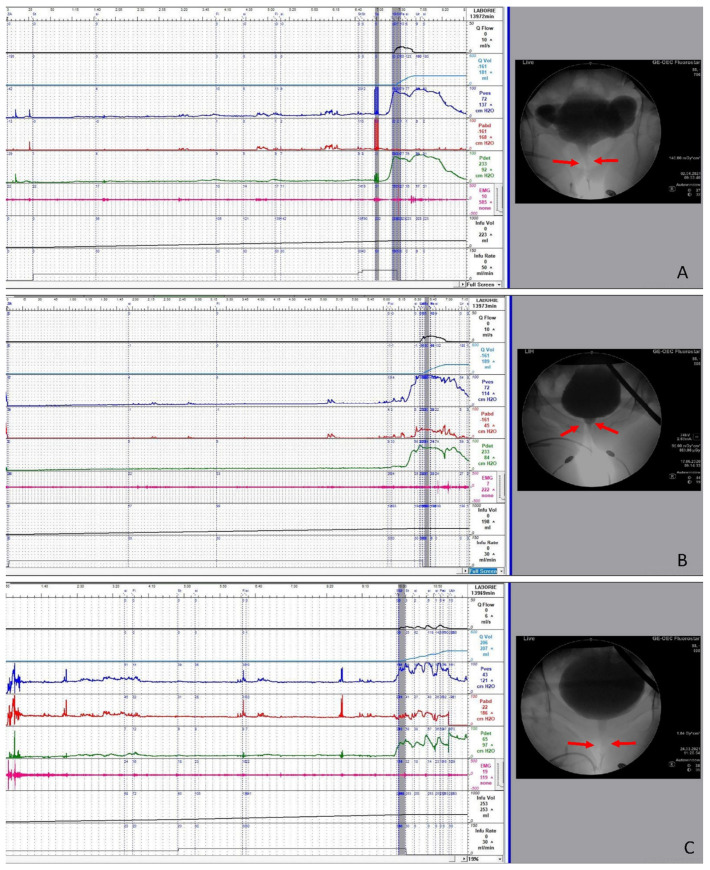
Videourodynamic tracings of female patients with dysfunctional voiding. Urethral narrowing was noted at (**A**) mid-urethra after previous transurethral bladder neck incision for bladder neck obstruction, (**B**) mid-urethra, and (**C**) distal urethra.

**Table 1 toxins-13-00362-t001:** Urodynamic parameters of female patients with dysfunctional voiding and successful or failed treatment outcome.

Parameter	Successful (n = 55)	Failure (n = 26)	*p*-Value
Age	59.7 ± 13.9	60.4 ± 19.3	0.857
FSF (mL)	146.6 ± 70.6	115.0 ± 52.8	0.046
FS (mL)	216.2 ± 86.4	184.9 ± 68.3	0.109
US (mL)	254.7 ± 99.3	215.5 ± 84.2	0.086
Pdet.Qmax (cmH_2_O)	59.0 ± 44.5	50.8 ± 35.2	0.414
Compliance	71.0 ± 66.6	58.9 ± 67.2	0.451
Qmax (mL/s)	8.06 ± 5.77	6.88 ± 4.09	0.352
Volume (mL)	158.3 ± 110.8	122.9 ± 78.7	0.104
PVR (mL)	179.8 ± 141.7	167.7 ± 101.9	0.698
CBC (mL)	338.1 ± 137.3	290.6 ± 105.1	0.123
VE	48.5 ± 32.5	44.8 ± 28.3	0.614
BOOI	42.8 ± 47.0	37.0 ± 35.7	0.578
DO	42(76.4%)	18 (69.2%)	0.629
DV subtype			0.376
BND + DV	4 (7.3%)	3 (11.5%)	
Mid-urethra DV	40 (72.7%)	15 (57.7%)	
Distal urethra DV	11 (20%)	8 (30.8%)	
CNS lesion	13 (24.1%)	2 (7.7%)	0.125
Storage LUTS	37 (67.3%)	20 (76.9%)	0.375
Voiding LUTS	49 (89.1%)	22 (84.6%)	0.719

FSF: First sensation of filling; FS: Full sensation; US: Urge sensation; Pdet.Qmax: Detrusor pressure at Qmax; Qmax: Maximum flow rate; PVR: Postvoid residual volume; CBC: Cystometric bladder capacity; VE: Voiding efficiency; BOOI: Bladder outlet obstruction index; DO: Detrusor overactivity; BND: Bladder neck dysfunction; DV: Dysfunctional voiding; CNS: Central nervous system; LUTS: Lower urinary tract symptoms.

**Table 2 toxins-13-00362-t002:** Urodynamic parameters and global response assessment of female patients with dysfunctional voiding and different videourodynamic characteristics.

Parameter	BND + DV (n = 7)	Mid-Urethra DV (n = 55)	Distal Urethra DV (n = 19)	*p* Value
Age	58.3 ± 20.5	57.1 ± 15.4	68.7 ± 11.6	0.018
FSF (mL)	140.9 ± 75.1	137.7 ± 72.1	131.3 ± 47.9	0.923
FS (mL)	215.7 ± 117.2	196.4 ± 80.5	230.7 ± 69.6	0.279
US (mL)	265.4 ± 109.1	226.4 ± 92.4	279.1 ± 94.5	0.093
Pdet.Qmax (cmH_2_O)	86.9 ± 60.6	61.8 ± 41.2	29.4 ± 14.3	0.001
Compliance	65.7 ± 55.7	61.7 ± 64.7	83.2 ± 75.9	0.483
Qmax (mL/s)	5.7 ± 4.0	7.9 ± 5.6	7.7 ± 4.8	0.588
Volume (mL)	98.1 ± 55.6	155.6 ± 112.4	139.8 ± 80.8	0.359
PVR (mL)	211.4 ± 159.9	160.0 ± 134.4	209.0 ± 98.4	0.277
CBC (mL)	309.6 ± 126.3	315.6 ± 129.1	348.8 ± 133.7	0.608
VE	0.407 ± 0.322	0.513 ± 0.337	0.384 ± 0.196	0.252
BOOI	75.4 ± 62.4	45.9 ± 43.7	13.9 ± 10.2	0.001
DO	6 (85.7%)	41 (74.5%)	9 (47.4%)	0.063
CNS lesion	0	11 (20.4%)	4 (21.1%)	0.539
Storage LUTS	5 (71.4%)	40 (72.7%)	12 (63.2%)	0.798
Voiding LUTS	6 (85.7%)	48 (87.3)	17 (89.5%)	1.000
Successful outcome	4 (57.1%)	40 (72.7%)	11 (57.9%)	0.376
GRA= +3	0	12 (21.8%)	3 (15.8%)	
GRA= +2	2 (28.6%)	18 (32.7%)	5 (26.3%)	
GRA= +1	2 (28.6%)	10 (18.2%)	3 (15.8%)	
GRA= 0	3 (42.9%)	12 (21.8%)	8 (42.1%)	
GRA= −1	0	3 (5.5%)	0	

BND: Bladder neck dysfunction; DV: Dysfunctional voiding; FSF: First sensation of filling; FS: Full sensation; US: Urge sensation; Pdet.Qmax: Detrusor pressure at Qmax; Qmax: Maximum flow rate; PVR: Postvoid residual volume; CBC: Cystometric bladder capacity; VE: Voiding efficiency; BOOI: Bladder outlet obstruction index; DO: Detrusor overactivity; CNS: Central nervous system; LUTS: Lower urinary tract symptoms.

**Table 3 toxins-13-00362-t003:** The lower urinary tract symptoms before and after urethral sphincter BoNT-A injection in patients with dysfunctional voiding.

	Total DV (n = 81)		BND + DV (n = 7)		Mid-Urethra DV (n = 55)		Distal Urethra DV (n = 19)	
	BL	FU	BL	FU	BL	FU	BL	FU
**Storage LUTS**	57 (70.4%)	41 (50.6%)	5 (71.4%)	3 (42.9%)	40 (72.7%)	30 (54.5%)	12 (63.2%)	8 (42.1%)
Frequency/urgency/nocturia	19	17	1	1	10	10	8	6
Urgency incontinence	38	24	4	2	30	20	4	2
**Voiding LUTS**	71 (87.7%)	34 (42.0%)	6 (85.7%)	2 (28.6%)	48 (87.3%)	21 (38.2%)	17 (89.5%)	11 (57.9%)
Difficult urination	67	34	5	2	45	21	17	11
Urinary retention	4	0	1	0	3	0	0	0
**Painful LUTS**	11 (13.6%)	4 (4.9%)	1 (14.3%)	0 (0%)	5 (9.1%)	2 (3.6%)	0	0
Bladder pain	6	2	1	0	5	2	0	0
Miction pain	5	2	2	1	3	1	0	0

DV: Dysfunctional voiding; BND: Bladder neck dysfunction; LUTS: Lower urinary tract symptoms; BL: Baseline; FU: Follow-up.

**Table 4 toxins-13-00362-t004:** Changes in urodynamic parameters after urethral botulinum toxin A treatment between the treatment success and failure groups.

Parameter		Successful (n = 55)	Failure (n = 26)	*p*
FSF (mL)	BL FU	152.3 ± 79.5 150.4 ± 94.3	119.2 ± 52.4 131.7 ± 105.6	0.652
FS (mL)	BL FU	225.2 ± 97.0 206.9 ± 102.8	195.4 ± 67.7 173.1 ± 125.9	0.910
US (mL)	BL FU	257.8 ± 107.6 232.1 ± 119.9	226.0 ± 82.6 192.8 ± 136.3	0.839
Pdet.Qmax (cmH_2_O)	BL FU	59.9 ± 38.8 44.4 ± 33.6 *	51.2 ± 36.0 36.2 ± 35.7	0.959
Compliance	BL FU	60.9 ± 52.8 64.2 ± 63.8	69.6 ± 84.7 57.3 ± 61.2	0.526
Qmax (mL/s)	BL FU	8.51 ± 6.21 8.83 ± 8.19	7.53 ± 4.72 4.85 ± 4.33	0.254
Volume (mL)	BL FU	146.2 ± 108.2 160.1 ± 140.7	126.5 ± 87.1 82.9 ± 73.4	0.185
PVR (mL)	BL FU	206.0 ± 137.5 132.6 ± 156.1 *	195.3 ± 101.6 243.3 ± 239.2	0.032
CBC (mL)	BL FU	352.2 ± 145.2 292.6 ± 136.9 *	321.9 ± 116.6 326.2 ± 215.4	0.185
Voiding efficiency	BL FU	42.7 ± 28.3 58.2 ± 36.8 *	40.3 ± 27.4 34.3 ± 32.4	0.057
BOOI	BL FU	42.9 ± 41.8 26.7 ± 39.5 *	36.1 ± 37.4 26.5 ± 32.8	0.630

* Significant difference between the baseline and follow-up data. FSF: First sensation of filling; FS: Full sensation; US: Urge sensation; Pdet: Detrusor pressure; Qmax: Maximum flow rate; PVR: Postvoid residual volume; CBC: Cystometric bladder capacity; BOOI: Bladder outlet obstruction index; BL: Baseline; FU: Follow-up.

## Data Availability

Data are available on request to the corresponding author.

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
