# Peer review of "Therapeutic Effects of Urethral Sphincter Botulinum Toxin A Injection on Dysfunctional Voiding with Different Videourodynamic Characteristics in Non-Neurogenic Women"

_toxins, 2021, doi:10.3390/toxins13050362_

Round 1

Reviewer 1 Report

The paper is interesting and well-written. I have only one remark regarding the results. It would be of great interest to know many of the patients successfully treated reported outcome as excellent(+3), markedly improved (+2) or mildly improved (+1). The authors should provide this data.

Author Response

Reply: Thank you for the comment. We have added the distribution of GRA after Botox injection among three subgroups, (lines 70-71) in the Results section and in Table 2.

Reviewer 2 Report

Dear authors,

Thank you for submitting this article describing an interesting topic.

However, I would suggest some improvements:

Abstract :

Line 5, please add is in the sentence “it difficult to treat”.

Line 9, the sentence “global response assessment by 1 or more” is hard to understand out of context.

Introduction

Could you please describe the epidemiology of the DV to emphasize the relevance of the article?

Line 35, I will highly recommend using the “complete” ICS definition

This is characterized by an intermittent and/or fluctuating flow rate due to involuntary intermittent contractions of the peri-urethral striated or levator muscles during voiding in neurologically normal women. This type of voiding may also be the result of an acontractile detrusor (abdominal voiding) with electromyography (EMG) or video-urodynamics required to distinguish between the two entities.

Reference

Haylen BT, de Ridder D, Freeman RM, Swift SE, Berghmans B, Lee J, Monga A, Petri E, Rizk D, Sand PK, Schaer GK An International Urogynecological Association (IUGA) / International Continence Society (ICS) joint report on the terminology for female pelvic floor dysfunction. Neurourol Urodyn, 2010,29:4-20; International Urogynecology J, 2010,21:5-26.

Results

A flow chart would have been appreciated.

You described the “global response assessment” in materials and methods, but you did not present these results. Could you please add them?

You did not describe the potential adverse effect.

Line 81, you described a decrease of Pdet. Could you please dichotomize a decrease of Pdet associated to DO during the filling phase and Pdet during the voiding phase?

Discussion

I would suggest improving your discussion plan, which is interesting but difficult to follow (i.e., discussing your main result with literature, pathophysiological hypotheses to explain your different patterns, the current treatment and the place of BoNT-A, and your study limitations).

Line 86-88, you wrote, “ Logistic regression of the urodynamic parameters, including DO subtypes, and clinical variables revealed that a greater first sensation of filling predicts a successful treatment outcome (p= 0.047)”. Thus, you described a predictive factor, which is the opposite of what you claimed in line 98 to 100.

I do agree with you; the involvement of the bladder neck is very interesting for pathophysiological hypothesis ( 1) line 131). It would be very interesting to dichotomize somatic and sympathetic reflexes.   

Materials and methods

The materials and methods is well described.

However, could you please justify (reference) your definition of DV (line 202)?

It would have been interesting to describe the lower urinary tract symptoms (voiding and storage) you analyzed before and after BoNT-A.

In your VUDS parameters, please add the full sensation and urge sensation as described in your results.

Could you please justify your successful treatment outcome?

Could you please add the logistic regression in your methodology?

Were patients treated with antimuscarinic or intradetrusor botulinum toxin during the study?

Author Response

Abstract :

Line 5, please add is in the sentence “it difficult to treat”

Reply: Thank you for the comment. We have added “is” on the sentence, accordingly. (line 5)

Line 9, the sentence “global response assessment by 1 or more” is hard to understand out of context.

Reply: Thank you for the comment. We have revised the statement to “reported global response assessment by ≥ 1”. (line 8)

Introduction

Could you please describe the epidemiology of the DV to emphasize the relevance of the article?

Reply: Thank you for the comment. We have added several statements to describe the epidemiology and clinical relevance of this article. (lines 25-28 and lines 29-32)

Line 35, I will highly recommend using the “complete” ICS definition

This is characterized by an intermittent and/or fluctuating flow rate due to involuntary intermittent contractions of the peri-urethral striated or levator muscles during voiding in neurologically normal women. This type of voiding may also be the result of an acontractile detrusor (abdominal voiding) with electromyography (EMG) or video-urodynamics required to distinguish between the two entities.

Reference

Haylen BT, de Ridder D, Freeman RM, Swift SE, Berghmans B, Lee J, Monga A, Petri E, Rizk D, Sand PK, Schaer GK An International Urogynecological Association (IUGA) / International Continence Society (ICS) joint report on the terminology for female pelvic floor dysfunction. Neurourol Urodyn, 2010,29:4-20; International Urogynecology J, 2010,21:5-26.

Reply: Thank you for the comment. The definition of DV has been revised according to the ICS definition. (lines 34-38)

Results

A flow chart would have been appreciated.

Reply: Thank you for the comment. Because this is a retrospective analysis of therapeutic effects of Botox on female DV, no flow chart for the treatment and follow-up is available.

You described the “global response assessment” in materials and methods, but you did not present these results. Could you please add them?

Reply: Thank you for the comment. The results of the global response assessment in female DV with different video urodynamic characteristics have been added in Table 2 and described in the Results section.

You did not describe the potential adverse effect.

Reply: Thank you for the comment. The adverse events after urethral Botox injection have been added to the Results section. (lines 70-71)

Line 81, you described a decrease of Pdet. Could you please dichotomize a decrease of Pdet associated to DO during the filling phase and Pdet during the voiding phase?

Reply: Thank you for the comment. The decrease of Pdet indicates voiding detrusor pressure during voiding phase, we have revised it to Pdet at Qmax (Pdet.Qmax). (line 69)  Regarding the Pdet of DO at filling phase, the Pdet showed no significant change after BoNT-A injection. (lines 93-94)

Discussion

I would suggest improving your discussion plan, which is interesting but difficult to follow (i.e., discussing your main result with literature, pathophysiological hypotheses to explain your different patterns, the current treatment and the place of BoNT-A, and your study limitations).

Reply: Thank you for the comment. We have revised the paragraphs in the Discussion section according to your suggestion and deleted some redundant statements in this section.

Line 86-88, you wrote, “ Logistic regression of the urodynamic parameters, including DO subtypes, and clinical variables revealed that a greater first sensation of filling predicts a successful treatment outcome (p= 0.047)”. Thus, you described a predictive factor, which is the opposite of what you claimed in line 98 to 100.

Reply: Thank you for the comment. The logistic regression tried to find predictive factors for a successful outcome of all DV patients. (lines 249-250)  In the first paragraph of Discussion section, we have revised the statement: “the study failed to find the difference of the successful treatment outcome among different VUDS characteristics of DV in this cohort.” (lines 112-113, and lines 171-172)

I do agree with you; the involvement of the bladder neck is very interesting for pathophysiological hypothesis ( 1) line 131). It would be very interesting to dichotomize somatic and sympathetic reflexes.   

Reply: Thank you for the comment. We have added the statement of different innervation reflex of the bladder neck and urethral sphincter. The increased sympathetic hyperactivity resulting in BN dysfunction and increased urethral smooth muscle hypertonicity might play a role in DV; however, urethral striated muscle spasm due to increased pudendal hyperactivity may also coexist. (lines 141-143)

Materials and methods

The materials and methods is well described. However, could you please justify (reference) your definition of DV (line 202)?

Reply: Thank you for the comment. The definition was made according to the recommendation of ICS/IUGA joint report. All patients included in this study had evident radiographic obstruction at the middle urethra with an open BN and characteristic urodynamic findings such as high Pdet.Qmax and low Qmax. (Lines 193-195)

It would have been interesting to describe the lower urinary tract symptoms (voiding and storage) you analyzed before and after BoNT-A.

Reply: Thank you for the comment. The lower urinary tract symptoms (LUTS) have been added in the Results section and tabulated as a new Table 3.  The lower urinary tract symptoms (LUTS) including storage (urgency/frequency/nocturia), voiding (difficult urination, urinary retention) and painful LUTS (bladder pain, micturition pain) were recorded before and 1 to 3 months after urethral BoNT-A injection. (lines 200-202, and lines 81-84)

In your VUDS parameters, please add the full sensation and urge sensation as described in your results.

Reply: Thank you for the comment. We have added fullness sensation, urge sensation in the text, accordingly. (line 215)

Could you please justify your successful treatment outcome?

Reply: Thank you for the comment. Patients with an improvement of VE by 10% and global response assessment by ≥ 1 were considered to have successful treatment outcome, otherwise the treatment was considered failure. (lines 240-242)

Could you please add the logistic regression in your methodology?

Reply: Thank you for the comment. We have added the logistic regression analysis in the Methods. (lines 249-250)

Were patients treated with antimuscarinic or intradetrusor botulinum toxin during the study.

Reply: Thank you for the comment. Patients who were taking antimuscarinics for the overactive bladder symptoms continued the medication after urethral BoNT-A injection. No concomitant intra-detrusor BoNT-A injection was performed. (lines 233-235)

Round 2

Reviewer 2 Report

Dear authors,

thank you for these improvements.

I would just suggest to correct the line 134 : "women with DV already had a 133wide BN open during voidinghad the greatyer improvedment of voiding LUTS"

Author Response

Reply: Thank you, the sentence had been revised, accordingly. In this study, women with DV already had a wide open BN during voiding had the greater improvement of voiding LUTS,